# Competency in ECG Interpretation and Arrhythmias Management among Critical Care Nurses in Saudi Arabia: A Cross Sectional Study

**DOI:** 10.3390/healthcare10122576

**Published:** 2022-12-19

**Authors:** Mohammed Saeed Aljohani

**Affiliations:** Medical-Surgical Nursing Department, Nursing College, Taibah University, Al-Madinah 42362, Saudi Arabia; msejohani@taibahu.edu.sa

**Keywords:** electrocardiography, interpretation, critical care nursing, competency

## Abstract

Background: Electrographic interpretation skills are important for healthcare practitioners caring for patients in need of cardiac assessment. Competency in ECG interpretation skills is critical to determine any abnormalities and initiate the appropriate care required. The purpose of the study was to determine the level of competence in electrocardiographic interpretation and knowledge in arrhythmia management of nurses in critical care settings. Methods: A descriptive cross-sectional design was used. A convenience sample of 255 critical care nurses from 4 hospitals in the Al-Madinah Region in Saudi Arabia was used. A questionnaire was designed containing a participant’s characteristics and 10 questions with electrocardiographic strips. A pilot test was carried out to evaluate the validity and reliability of the questionnaire. Descriptive and bivariate analyses were conducted using an independent t-test, one-way ANOVA, or bi-variate correlation tests, as appropriate. A statistical significance of *p* < 0.05 was assumed. Results: Females comprised 87.5% of the sample, and the mean age of the sample was 32.1 (SD = 5.37) years. The majority of the participants (94.9%) had taken electrocardiographic interpretation training courses. The mean total score of correct answers of all 10 ECG strips was 6.45 (±2.54) for ECG interpretation and 4.76 (±2.52) for arrhythmia management. No significant differences were observed between ECG competency level and nursing experience or previous training. Nurses working in the ICU and CCU scored significantly higher than those working in ED. Conclusions: The electrocardiographic knowledge in ECG interpretation and arrhythmia management of critical care nurses is low. Therefore, improving critical care nurses’ knowledge of ECGs, identification, and management of cardiac arrhythmias is essential.

## 1. Introduction

The prevalence of cardiac arrhythmias, electrical heart conduction system diseases, and other cardiovascular diseases (CVD), in general, is increasing worldwide [1,2,3]. Cardiac arrhythmias are defined as a disturbance in the normal heart electrical conduction system, resulting in ineffective cardiac pumping, unstable hemodynamic, or cardiac arrest events [2]. Cardiac arrhythmia is one of the leading causes of death globally. In 2016, the World Health Organization (WHO) estimated that 31% (17.9 million) of all global deaths were caused by CVDs [4]. In Saudi Arabia, the WHO and the Ministry of Health (MOH) Statistical Yearbook revealed that cardiovascular diseases were responsible for 42% of non-communicable disease deaths in 2010 [5].

Electrocardiogram (ECG) is a valuable non-invasive diagnostic tool for rapid identification of many heart diseases, especially electrico-cardiac arrhythmias and acute coronary syndrome [6,7,8]. ECG monitoring is commonly indicated for patients who have a risk of arrhythmias or suspected ischemic heart disease [9,10]. Nurses play a critical role in providing care in critical care settings such as the emergency department (ED), intensive care unit (ICU), and cardiac care unit (CCU) [11,12]. Usually, patients in these departments require ECG monitoring.

Thus, they are required to have sufficient knowledge and skills to provide comprehensive and safe healthcare for all patients with different cardiac diseases, particularly the critically ill in hospitals [13,14].

Nurses usually are the first clinicians to look at the ECG results and to identify abnormalities in the ECG which may require immediate attention. Therefore, it is vital that nurses are competent to carry out an initial assessment and make an early identification and quick decisions to manage ECG abnormalities and activate appropriate emergency health teams or initiate first-line treatments [15,16,17]. Nurses’ rapid and accurate interpretation of cardiac arrhythmias has been linked to safe practices and positive patient outcomes [18,19,20,21,22].

No consensus exists in the literature about the meaning of competency in ECG interpretation and the cut-off point for competency. However, it has been stated that competency can be defined as the ability to have sound understanding of the theoretical and procedural knowledge to interpret cardiac rhythms (knowledge), the ability to recognize cardiac rhythms (skills), and possession of a reasonable level of confidence to effectively undertake the task (attitude) [19].

Several studies conducted worldwide have reported different levels of nurse competency in arrhythmia interpretation and management [10,23,24,25,26,27,28]. For example, one study reported low competency in ECG interpretation among emergency nurses [29]. In a Turkish study, a high proportion (61%) of bedside nurses reported they did not know the correct practice for ECG monitoring or the correct interpretation of arrhythmias [24]. Another study, conducted in Iraq to investigate nurses’ knowledge of early interventional treatment for patients with ventricular tachycardia, showed nurses lack knowledge of how to interpret an ECG and recognize ventricular tachycardia arrhythmias [27]. However, other studies reported high competency scores in ECG interpretation. For example, a 2017 study reported ECG knowledge was high among ED nurses [23]. They also found that knowledge was influenced by ECG training in the previous five years but was not influenced by work experience or the hospital type.

In relation to nurses’ knowledge about early intervention for arrhythmia, two previous studies reported a low level of nurses’ knowledge regarding management of life-threatening ventricular arrhythmias [26,27]. However, there is a scarcity of studies in Saudi Arabia evaluating nurses’ competency in the interpretation of ECG. In fact, only one study was published in this area in 2022 [30]. That study sought to identify nurses’ competencies in ECG interpretation. The study focused on their ability to identify cardiac arrhythmia; however, it did not investigate knowledge of proper/initial management.

Therefore, this study aims to identify the level of competency in ECG interpretation, including the ability to identify arrhythmia and its initial management among nurses working in critical care settings in the Al-Madinah region, Saudi Arabia. It is expected that the findings of this study will be beneficial to understand the current nurses’ competency in ECG interpretation and arrhythmias management. Furthermore, it is expected to help in establishing nursing programs that enhance nurses’ knowledge and skills in relation to ECG interpretation and arrhythmias management.

### Study Objectives

Specific objectives of this study were to identify: (1) critical care nurses’ competency levels in ECG interpretation and arrhythmia management, (2) relationship between critical care nurses’ ECG interpretation and arrhythmia management, and (3) the relationship between participants’ demographic and work data and competency level in ECG interpretation and arrhythmia management.

## 2. Materials and Methods

### 2.1. Study Design

A descriptive cross-sectional design was used to determine critical care nurses’ knowledge about common cardiac arrhythmia interpretation and management.

### 2.2. Sample and Settings

A convenience sampling method was used to recruit critical care nurses working in critical care units at four main governmental hospitals in two large cities in the Al-Madinah region in Saudi Arabia. These four central hospitals provide a wide range of care for patients with different medical disorders, including cardiac diseases.

To address the objectives of this study, nurses from the selected hospitals who, at the time of data collection, were working in critical care settings including the intensive care unit (ICU), coronary care unit (CCU), and emergency department (ED) were invited to participate. Excluded were nurses who do not provide direct care or are not currently working in critical care units.

The estimated sample size was calculated using G* power software [31]. The effective sample size was determined according to the type of analysis (correlation and Chi square analysis), a medium effect size of (0.3), the power analysis level of (0.80), and a significant *p* value of (0.05). Based on previous data, the minimal effective sample size was calculated to range between 158 (for correlation analysis) and 221 nurses (for Chi square analysis). A total of 270 questionnaires were returned, and 15 questionnaires were excluded for being incomplete; 255 completed questionnaires were collected and used for the final analysis of this study.

### 2.3. Procedure

The researcher visited the selected hospitals to recruit study participants. The researcher approached head nurses of the critical care units in each hospital to explain the purpose of this study and to disseminate the link to the online questionnaire, through email or WhatsApp, to the participants. The online questionnaire included an explanation of the objectives of the study and provided instructions on how to complete the questionnaire that was used for data collection. Printed versions of the questionnaire were also provided to the head nurses to promote participation and to increase the response rate. The online survey was hosted on a special webpage created expressly for this purpose. All replies were made to this webpage then extracted to a SPSS file for analysis.

### 2.4. Instrument

The goal of this study was to understand the competence in interpreting an ECG and the knowledge of first line management for selected arrhythmias. A structured questionnaire was developed and utilized for collecting data to address the objectives of this study. The questionnaire consists of two parts. Part 1 elicited demographic data (age, gender, nationality, qualification, hospital, working area experience, electrocardiographic training, and type of training). Part 2 presented 10 lead II ECG strips. The selected ECG strips were all made with the same technique (tracing speed and calibration, ECG voltage, and same paper) to prevent any technical variations or negatively influence the recognition of arrhythmias. For each strip, participants were asked to answer three questions. No. 1: participants were requested to identify (interpret) the displayed rhythm by selecting one answer only out of eleven possible answers. No. 2: participants were asked to rate the difficulty identifying the cardiac arrhythmia on a three-level scale (1 = easy, 2 = moderate, 3 = difficult). No. 3: participants were asked to identify the initial (first line) management of the cardiac arrhythmia shown in the ECG strip by selecting one answer only out of four possible answers (Table 1). The questions were developed by the author based on reviewing different resources, including textbooks related to ECG interpretation and management of cardiac arrhythmias [32,33,34], and guidelines of basic life support, cardiopulmonary resuscitation, and advance cardiac life support published by the American Heart Association [35,36].

The maximum score for interpretation and management questions was 10 points, with each corrected question given 1 point. Participants who scored at least 7.5 out of 10 points were deemed competent in electrocardiographic interpretation and/or management of patients with types of CVD [23]. Conversely, those who scored less than 7.5 points were not considered competent.

To assess the clarity, readability, and reliability of the developed questionnaire, a pilot study was conducted by recruiting 20 nurses from 2 hospitals not included in the final study. Minor revisions were made based on written feedback received from the piloted nurses. The internal reliability of the developed questionnaire was good (Cronbach’s alpha = 0.85). Two PhD holders in critical nursing and two ICU consultants, who are experts in clinical health research and cardiac arrhythmia interpretations and management, reviewed the questionnaire to assess the validity, clarity, and feasibility of the developed questionnaire. Some modifications were made to the tool based on expert suggestions.

### 2.5. Ethical Considerations

Institutional Review Board approval was obtained from the General Directorate of Health Affairs in Madinah (H-03-M-084) before the commencement of this study. Participants were informed that participation is voluntary, names are not required, and they can withdraw from the study at any time. Informed consent was assumed if participants returned a completed questionnaire. Data were confidential; no one, except the investigator, had the authority to view the data. No names were sought; instead, an ID number was assigned to each questionnaire.

### 2.6. Data Analysis

Statistical Package of Social Science (SPSS) software program (version 22) was used to analyze the data. Descriptive statistics were used to describe participants’ characteristics. Chi square tests were utilized to identify the relationship between those who correctly interpreted the ECG strips and correctly identified the management of cardiac arrhythmia in each ECG strip. Independent sample t-test, one way ANOVA, and bi-variate correlation tests were conducted, as appropriate, to determine the association between demographical and work data, participants’ knowledge about ECG arrhythmia interpretation, and management. A *p* value of less than 0.05 was considered a significant value.

## 3. Results

### 3.1. The Participants’ Sociodemographic

The majority of participants were female (87.5%, *n* = 223) and non-Saudi citizens (63.9%, *n* = 163). The mean age of participants was 32.1 years (SD = 5.37). Two-thirds of nurses (75.7%, *n* = 193) had Bachelor’s degrees and three years or less of experience (44.7%). More than half of the participants were working at ICUs (59.2%, *n* = 151). Most had taken training courses related to cardiac rhythm interpretation or arrhythmia management (94.9%), and 63.9% had completed at minimum the basic life support (BLS) course (Table 2).

### 3.2. Knowledge in ECG Interpretation

Participants’ competency on ECG interpretation knowledge was low. As shown in Table 2, the mean total score of correct answers of all 10 ECG strips was 6.45 (±2.54) (out of 10). Only 38 participants (14.9%) recognized all 10 ECG arrhythmias correctly, while almost one-fourth (27.1%) of participants answered 8 or more questions correctly, and 45.5% of participants scored 6 correct answers or less. The most frequently identified items (94.1% and 75.7%, respectively) were ECG strip No. 5 (asystole) and ECG strip No. 2 (ventricular tachycardia). Additionally, about two-thirds of participants were correctly able to identify the interpretation of ECG strip No. 7 (sinus tachycardia) (65.1%), and ECG strip No. 10, pulseless electrical activity (PEA) (63.1%). Moreover, about half of the participants were correctly able to identify the interpretation of ECG strip No. 3, ventricular fibrillation (53.7%), ECG strip No. 4, atrial flutter (58.4%), ECG strip No. 8, atrial fibrillation (50.2%), and ECG strip No. 9, third degree complete heart block (56.9%). Almost 30% of participants failed to identify a normal sinus rhythm.

Furthermore, participants were asked about the difficulty in identifying the ECG rhythm based on three levels (1 = easy, 2 = moderate, 3 = difficult). The majority (64.9%) perceived the interpretation of the provided ECG cases as easy, the total mean score was 1.7 (±0.5). Of all participants, 81.2% said ECG strip No. 5 (asystole) was easy to interpret. Moreover, 30.2% said ECG strip No. 3 (ventricular fibrillation) was difficult to interpret (Table 3).

### 3.3. Knowledge of Arrhythmia Management

Participants’ competency on first line management of ECG arrhythmias was low. The mean total score of correct answers of all 10 ECGs was 4.76 (±2.52) (out of 10). No one correctly identified all 10 ECG arrhythmias, while 61.6% of participants identified 6 correct answers or less. The most frequently known and correctly identified management (79.6%) was ECG strip No. 5 (asystole). As seen in Table 4, only 32.9% identified the correct first line management for ECG strip No. 8 (atrial fibrillation).

As shown in Table 5, Chi square tests show significant relationships between correct interpretation of all ECG strips and correctly identifying the management for each ECG strip. In other words, participants who were more knowledgeable about interpretation of ECG strips were more knowledgeable about the management of cardiac arrhythmias (Table 4).

### 3.4. Factors Associated with ECG Interpretation and Arrhythmias Management Knowledge

Several variables were examined for an association between ECG interpretation and the knowledge of arrhythmias management score. Using the independent samples t-test, the nationality (Saudi and non-Saudi) was found to have a significant effect on the ECG interpretation and arrhythmias management knowledge score (t(253) = 2.022, *p =* 0.044) for interpretation and (t(253) = 2.978, *p =* 0.003) for management. The mean score for non-Saudi nurses (M = 6.78, SD = 2.51 and M = 5.11, SD = 2.49) was slightly higher (Table 5) compared to the Saudi nurses (M = 6.11, SD = 2.53 and M = 4.15, SD = 2.46).

Regarding the association between what unit the nurse worked on with the ECG interpretation and arrhythmias management knowledge score, results showed a statistically significant difference between groups as determined by one-way ANOVA (F(3) = 6.80, *p* = 0.01 for interpretation, and F(2) = 12.67, *p* = 0.001 for management. A Tukey post hoc test revealed the mean score for nurses in the ED (M = 5.42 and 3.48) was statistically significantly lower than those working in the ICU (M = 6.99, 5.25, *p* = 0.01) and CCU (M = 6.71 and 5.07, *p* = 0.001). There was no statistically significant difference between the ICU and CCU mean score (*p* = 0.918).

Additionally, a Pearson correlation coefficient was computed to assess the relationship between the participants’ perceived difficulty level and the ECG interpretation and arrhythmias management knowledge. There was a moderate negative correlation between the two variables, r = −0.58 and −0.55, *p* ≤ 0.00. In other words, participants who rated interpretation of the ECG as difficult were more likely to obtain low scores in both ECG interpretation and arrhythmia management.

Regarding other factors studied, results show there were no statistically significant differences, at the *p* < 0.05 level, between the ECG interpretation and arrhythmias management knowledge scores and age (*p* = 0.35) for interpretation and (*p* = 0.23) for management, gender (*p* = 0.85) for interpretation and (*p* = 0.58) for management, previous training (*p* = 0.18) for interpretation and (*p* = 0.25) for management, qualifications (*p* = 0.51) for interpretation and (*p* = 0.21) management, or work experience (*p* = 0.47) for interpretation and (*p* = 0.42) for management, (Table 6).

## 4. Discussion

The aim of this study was to identify nurses’ competency in ECG interpretation and arrhythmia management in critical care settings in four hospitals in the Al-Madinah region in Saudi Arabia. To the best of my knowledge, this is the first study to evaluate critical care nurses’ competency in ECG interpretation and arrythmias management in Saudi Arabia.

The competency level in this study was set at 7.5 out of 10. Results of the study showed overall ECG interpretation among nurses working in critical care settings (ICU, CCU and ED) was below the preset threshold (7.5 out of 10). The mean total score of correct answers of all 10 ECGs was 6.45 (±2.54). This finding is consistent with the results of other local and international studies [29,30,37]. A recent study in Saudi Arabia found that 50% of nurses working in the ICU and CCU showed low competency in ECG interpretations [30]. The total mean score was 6.68 out of 10. However, the competency level in the present study is lower than in other studies [23,38]. When Coll-Badell et al. evaluated the knowledge of nurses in the ED, they found competency was high, where 93% of nurses scored 7.5 or more, with the average score of 8.6 out of 10 [23].

In the current study, only 14.9% of participants answered all 10 questions correctly. This finding is similar to Ho et al., that found 12.5% of participants correctly answered all the questions [38].

The findings of the present study showed the questions that were most often answered correctly were related to asystole (94%) and ventricular tachycardia (75.7%). However, one study, in contrast, found nurses’ ability to identify ventricular tachycardia was low (22%) [39]. The majority of participants in this study (81.2%) perceived the interpretation of the ECG involving asystole as easy. Given that most of the nurses in this study had completed BLS (63.9%) or ACLS (48.6%) courses, it is not surprising they can identify the asystole rhythm correctly and know how it should be managed. This finding is congruent with other studies [29,38,40]. In addition, the current study found nurses had difficulties identifying atrial flutter (58.4%), atrial fibrillation (50.2%), and third-degree complete heart block (56.9%). This finding was also similar in other studies [28,30,41]. In contrast, Tahboub and Dal Yılmaz found that 84.6% of nurses were able to identify atrial flutter [42]. In addition, about two-thirds of participants were able to interpret ECG strips related to sinus bradycardia, sinus tachycardia, and PEA. It can be argued that nurses need to recognize the normal ECG first to be able to identify any abnormalities. However, the findings showed that one-third of nurses in the current study had difficulty recognizing normal sinus rhythm. This finding was also reported in 2022, which found that 42% of the nurses failed to identify the normal sinus rhythm [30].

The second objective of the current study was to evaluate the competency level of nurses working in the critical care setting to identify first line management for some common fatal arrythmias. The results showed overall competency was below the required limits. The mean total score of correct first line management was 4.76 (±2.52) out of 10. This finding is similar to the findings of two previous studies which revealed the majority of the nurses had unsatisfactory knowledge concerning early management of life-threatening ventricular arrhythmias [26,27]. In the current study, no participant identified the correct first line management for all 10 ECG arrhythmias, while 61.6% identified 6 correct answers or less. The majority of participants (79.6%) identified first line management for asystole, while more than half had difficulties identifying the correct first line management for pulseless electrical activity, third degree complete heart block, atrial fibrillation, and ventricular fibrillation. In more detail, despite the fact more than 50% of the nurses correctly identified the ECG strips for ventricular fibrillation, atrial flutter, atrial fibrillation, and third-degree heart block, less than two-thirds were able to recognize the proper management of these cardiac arrhythmias. Moreover, between 81% and 85.5% of nurses in this study correctly recognized the management of cardiac arrhythmias for sinus bradycardia, ventricular tachycardia, and sinus tachycardia.

The findings of this study showed that there are significant relationships between participants’ knowledge about interpretation of ECG strip scenarios and management of ECG arrhythmias. These expected results indicated that nurses with a strong knowledge of arrhythmia interpretation were more knowledgeable about correct management of the cardiac arrhythmias and more likely to make sound clinical decisions and take quick interventional actions to manage cardiac arrhythmias [43,44,45]. Moreover, these findings emphasize the importance for improving critical care nurses’ knowledge about interpretation of ECG and early identification of cardiac arrhythmias and correct management of cardiac arrhythmias [44,45,46,47].

The finding of this study showed a significant association among participants’ accurate knowledge about the interpretation of ECG rhythms, participants’ knowledge about management of cardiac arrhythmias, and the nurses’ nationality. Non-Saudi nurses scored slightly higher than Saudi nurses. Although this finding is difficult to explain, it may be attributed to the fact there were 163 non-Saudi participants in this study and 92 Saudi participants. However, this finding may indicate a lack of knowledge of ECG interpretation and arrhythmias management among Saudi nurses. Therefore, ECG interpretation knowledge and skills must be addressed in Saudi nursing curricula and continuing education programs.

In addition, the current study found an association between ECG interpretation and arrhythmia management knowledge and the department nurses worked in. Mean score of nurses working in the ICU and CCU was significantly higher than those working in the ED. A similar study found CCU experience was associated with better results on ECG interpretation [37]. Another study revealed that nurses working in the CCU had better ECG knowledge than nurses working in the ICU and ED [48]. Contrary to this current research, a 2021 study found that nurses who had ED experience scored significantly higher than those who did not [38].

Several studies reported improved competency in ECG interpretation of fatal arrhythmias after education [9,25,39]. Logically, nurses with training showed higher scores compared to those without training [23,29,48]. It has been argued that knowledge and skills may diminish with time; therefore, refresher courses are required. Nolan and Coll-Badell et al. recommend that nurses take ECG interpretation courses at least every five years [23,49]. However, the findings of this current study are not consistent with previous research. The current study showed the mean score of participants was not affected by whether or not they had training. Although the majority of participants claimed to have training related to arrhythmia, it was not clear when training occurred or how affective it was. This finding draws attention to the need to evaluate arrhythmia training in both nursing curricula and on-the-job courses.

Data from the current study provided no significant difference among participants’ gender, age, qualifications, or work experience and their expertise in ECG interpretation. This result is consistent with two studies that found no correlation between age, work experience, and interpretation knowledge and skills [37,50]. In contrast, Keller et al. reported a positive correlation between years of experience and higher scores [40].

### 4.1. Research Implications and Recommendations for Clinical Practice

This study provides baseline information to improve the evidence about the level of knowledge of critical care nurses regarding ECG interpretation and cardiac arrhythmia management in Saudi Arabia. Moreover, this study will provide helpful information to educators to develop clinical guidelines and education programs to improve critical care nurses’ knowledge and competencies for early identification, assessment, and appropriate management of patients with CVDs, particularly cardiac arrhythmias and CAD.

It is essential to improve critical care nurses’ knowledge about monitoring ECGs and identifying cardiac arrhythmias. Several strategies are recommended to achieve this. ECG monitors with automated interpretation are highly recommended in critical care settings to facilitate early detection of abnormal ECG strips and diagnosis of cardiac arrhythmias. Most importantly, the curricula of Bachelor of Science in Nursing programs in Saudi Arabia should be reviewed and adjusted to include topics about ECG interpretations and management of cardiac arrhythmias.

### 4.2. Limitations

Despite the importance of the findings of this study, there were a few limitations. Using a cross-sectional design will not help identify the effect and causal relationship of the lack of knowledge on interpreting ECG and cardiac arrhythmias. Additionally, using convenience sampling will negatively affect the generalization of the findings for all critical care nurses in Saudi Arabia. This study was limited to only 10 ECG rhythms; other important arrythmias can be included in other studies, such as ST elevation or pathological Q waves.

## 5. Conclusions

This is the first study in Saudi Arabia to evaluate both ECG interpretation and arrhythmia management knowledge for nurses working in critical care settings. The overall results revealed the majority of nurses in this study were below the preset competency limit for ECG interpretation and arrhythmia management. Therefore, improving critical care nurses’ knowledge on monitoring ECGs and identification and management of cardiac arrhythmias is essential. Through collaboration between the health system and education institutions, improvements can be achieved through nursing education and in-service training programs and workshops. Contrary to what several studies found related to the role of training in improving nurses’ interpretation knowledge, this current study did not find any significant association. This finding should be carefully examined, as it may indicate weaknesses in current nursing curricula and/or in-service training programs. Conducting further longitudinal and experimental research studies is recommended to investigate the effectiveness of health education programs on critical care nurses’ ECG interpretation skills and management of cardiac arrhythmias.

## Figures and Tables

**Table 1 healthcare-10-02576-t001:** Example of the questionnaire questions.

Question No 1	**For each of the following ECG strips, please select the correct rhythm interpretation and also rate the level of difficulty that was required to interpret each strip on a scale of 1 to 3**
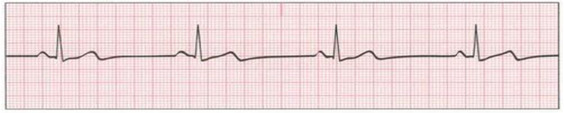
A.The above rhythm is:
□Ventricular Tachycardia	□Sinus Bradycardia
□First Degree Heart Block	□Atrial Fibrillation
□Asystole	□Pulseless Electrical Activity (PEA)
□Atrial Flutter	□Normal Sinus Rhythm
□Ventricular Fibrillation	□Sinus Tachycardia
□Third Degree (Complete) Heart Block)	
B.Difficulty level: 1.Easy2.Moderate3.Difficult
C.The above patient has hypotension and dizziness what is initial management
□Administer Atropine up to 3 mg while awaiting pacer	□No intervention is required
□Start CPR	□Administer Lidocaine 10 mg IV
Question No 2	2
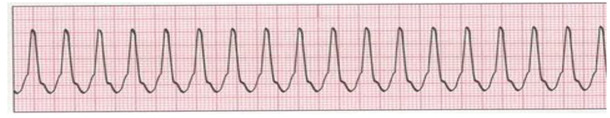
A.The above rhythm is:
□Ventricular Tachycardia	□Sinus Bradycardia
□First Degree Heart Block	□Atrial Fibrillation
□Asystole	□Pulseless Electrical Activity (PEA)
□Atrial Flutter	□Normal Sinus Rhythm
□Ventricular Fibrillation	□Sinus Tachycardia
□Third Degree Complete Heart Block)	
B.Difficulty level: 1.Easy2.Moderate3.Difficult
C.You checked the above patient and you did not find a pulse, what is initial management?
□Immediate CPR with rapid defibrillation	□Immediate synchronized cardioversion
□Administer Lidocaine 10 mg IV	□Atropine up to 3 mg IV while awaiting pacer

**Table 2 healthcare-10-02576-t002:** Sociodemographic of the study population.

Variable		*n*	%
Gender			
Male		32	12.5%
Female		223	87.5%
Nationality			
Saudi		92	36.1%
Non-Saudi		163	63.9%
Educational qualification			
Nursing Diploma		53	20.8%
Bachelor		193	75.7%
Postgraduate (Diploma, Master, or PhD)		9	3.5%
Current working area			
Intensive care unit (ICU)		151	59.2%
Coronary care unit (CCU)		38	14.9%
Emergency department		66	25.9%
Working experience in the current area			
Less than one year		29	11.4%
1 to 3 years		85	33.3%
4 to 6 years		46	18.0%
7 to 10 years		46	18.0%
More than 10 years		49	19.2%
Did you attend any training in cardiac rhythm interpretation or arrhythmia management?			
Yes		242	94.9%
No		13	5.1%
What type of training attended? *			
ECG interpretation		133	52.1%
Basic life support (BLS)		163	63.9%
Advance cardiac life support		124	48.6%
	Mean (±SD)	Median	Min.–Max.
Age (years)	32.1 (±5.37)	30	23–52

* The participants attended more than one training program or course.

**Table 3 healthcare-10-02576-t003:** Distribution by percentage of nurses’ ECG interpretation results.

Participants’ Interpretation of ECG Strips	Correct Answer		Participants’ Response	
			Correct *n* (%)	Incorrect *n* (%)
Interpretation of ECG 1	sinus bradycardia		173 (67.8%)	82 (32.2%)
Interpretation of ECG 2	ventricular tachycardia		193 (75.7%)	62 (24.3%)
Interpretation of ECG 3	ventricular fibrillation		137 (53.7%)	118 (46.3%)
Interpretation of ECG 4	atrial flutter		149 (58.4%)	106 (41.6%)
Interpretation of ECG 5	asystole		240 (94.1%)	15 (5.9%)
Interpretation of ECG 6	normal sinus rhythm		177 (69.4%)	78 (30.6%)
Interpretation of ECG 7	sinus tachycardia		166 (65.1%)	89 (34.9%)
Interpretation of ECG 8	atrial fibrillation		128 (50.2%)	127 (49.8%)
Interpretation of ECG 9	third degree complete heart block		145 (56.9%)	110 (43.1%)
Interpretation of ECG 10	pulseless electrical activity		161 (63.1%)	94 (36.9%)
	Mean (±SD)	Median	Minimum–maximum	
Total score of correct interpretation of all 10 ECG strips	6.45 (±2.54)	7.0	1–10	
Total mean score of perceived difficulties in interpretation of all 10 ECG strips	1.70 (±0.51)	1.7	1–3	

**Table 4 healthcare-10-02576-t004:** Nurses’ arrhythmia management results by percentage.

ECG Strip Number (Patient’s Condition)	Correct Answer	Participants’ Response	
		Correct *n* (%)	Incorrect *n* (%)
ECG 1: Sinus Bradycardia (The patient has hypotension and dizziness)	Administer atropine up to 3 mg while awaiting pacer	117 (45.9%)	138 (54.1%)
ECG 2: Ventricular Tachycardia (You checked the above patient and did not find a pulse)	Immediate CPR with rapid defibrillation	141 (55.3%)	114 (44.7%)
ECG 3: Ventricular Fibrillation	Immediate CPR with rapid defibrillation	125 (49%)	130 (51%)
ECG 4: Atrial Flutter (This patient is hemodynamically unstable)	Synchronized electrical cardioversion	138 (54.1%)	117 (45.9%)
ECG 5: Asystole	Immediate CPR and epinephrine 1 mg IV bolus every 3–5 min	203 (79.6%)	52 (20.4%)
ECG 6: Normal Sinus Rhythm	No intervention required	160 (62.7%)	95 (37.3%)
ECG 7: Sinus Tachycardia (This patient has a pulse but is hemodynamically unstable)	Perform immediate synchronized cardioversion	142 (55.7%)	113 (44.3%)
ECG 8: Atrial Fibrillation (This patient is hemodynamical unstable)	Perform immediate synchronized cardioversion	84 (32.9%)	171 (67.1%)
ECG 9: Complete Third-Degree Heart Block	Administer atropine and perform temporary pacemaker	118 (46.3%)	137 (53.7%)
ECG 10: Pulseless Electrical Activity (PEA)	CPR along with epinephrine	127 (49.8%)	128 (50.2%)
	Mean (±SD)	Median	Minimum-maximum
Total score of correct management of all 10 ECG strips	4.76 (±2.52)	5.0	1–10

**Table 5 healthcare-10-02576-t005:** Relationship between participants’ ECG interpretation knowledge and arrhythmias management.

ECG Strip No.	Total Correct Management	Incorrect ECG Interpretation and Correct Management	Correct ECG Interpretation and Correct Management	Chi Square (X2)	*p* Value
ECG 1	137 (53.7%)	20 (14.6%)	117 (85.4%)	41.8	<0.001
ECG 2	174 (68.2%)	33 (19.0%)	141 (81.0%)	8.5	0.005
ECG 3	204 (80.0%)	79 (38.7%)	125 (61.3%)	23.4	<0.001
ECG 4	221 (86.7%)	83 (37.6%)	138 (62.4%)	10.98	0.001
ECG 5	210 (82.4%)	7 (3.3%)	203 (96.7%)	13.97	0.001
ECG 6	180 (70.6%)	20 (11.1%)	160 (88.9%)	109.35	<0.001
ECG 7	166 (65.1%)	24 (14.5%)	142 (85.5%)	171.8	<0.001
ECG 8	138 (54.1%)	54 (39.1%)	84 (60.9%)	13.71	<0.001
ECG 9	184 (72.2%)	66 (35.9%)	118 (64.1%)	14.23	<0.001
ECG 10	139 (54.5%)	12 (8.6%)	127 (91.4%)	104.62	<0.001

**Table 6 healthcare-10-02576-t006:** Correlation between participants’ demographic characteristics, knowledge about management of ECG arrhythmias, and participants’ knowledge about interpretation of ECG strip scenarios.

Variables	Participants’ Knowledge about Interpretation of ECG Rhythm		Participants’ Knowledge about Management of Cardiac Arrhythmias	
	Correlation (r)	*p* Value	Correlation (r)	*p* Value
Age (years)	0.06	0.35	0.08	0.23
Gender (female)	0.01	0.88	0.04	0.56
Nationality (non-Saudi)	0.12	0.051	0.18	0.004
Educational qualification	0.16	0.01	0.19	0.003
Working experience in the current area	−0.04	0.48	−0.04	0.58
Attendance of training program/course about cardiac rhythm interpretation or arrhythmia management	0.17	0.03	0.23	0.02
Perceived difficulties in interpretation of all 10 ECG strips (Total score out of 30)	−0.58	<0.001	−0.55	<0.001
Participants’ knowledge about management of ECG arrhythmias			0.93	<0.001

## Data Availability

The data presented in this study are available on reasonable request from the author. The data are not publicly available due to privacy restrictions.

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
