# Peer review of "Competency in ECG Interpretation and Arrhythmias Management among Critical Care Nurses in Saudi Arabia: A Cross Sectional Study"

_healthcare, 2022, doi:10.3390/healthcare10122576_

Round 1
Reviewer 1 Report
This article studies how nurses are able to interpret ECGs, especially arrhythmias, as well as their initial management.
I have some comments:
11) L110: “The estimated sample size was calculated using G* power software [31]. The effective sample size was determined according to the type of analysis (correlation and Chi square analysis), a medium effect size of (0.3), the power analysis level of (0.80) and significant value of (0.05). Based on previous data, the minimal effective sample size was calculated to range between 158 (for correlation analysis) and 221 nurses (for Chi square analysis). “
Could you explain the equation you have used to determine the sample size? You could also explain the concepts “medium effect size” and ‘power analysis level”.
22) Section 2.4: could you show some parts of the questionnaire? Some examples of the most relevant parts?
33) L243:” Regarding other factors studied, results show there were no statistically significant differences, at the p<0.05 level, between the ECG interpretation and arrhythmias manage- 244 ment knowledge scores and age (p=0.35, p= 0.23), gender (p=0.85, p= 0.58), previous train- 245 ing (p=0.18,p= 0.25 ), qualifications (p=0.51,p= 0.21 ) or work experience (p=0.47,p= 0.42 ), 246 (Table 5). ><0.05 level, between the ECG interpretation and arrhythmias management knowledge scores and age (p=0.35, p= 0.23), gender (p=0.85, p= 0.58), previous training (p=0.18,p= 0.25 ), qualifications (p=0.51,p= 0.21 ) or work experience (p=0.47,p= 0.42 ), (Table 5).
Could you explain better this paragraph? It is confusing to find 2 “p” in parentheses. You could explain the “p” of ECG interpretation and then the “p” of arrhythimias management.
Reviewer 2 Report
In this cross-sectional study, the authors report on the competency of EKG interpretation among critical care nurses in health center in Saudi Arabia.
The research is well done and manuscript is clearly presented with appropriate methodology, results and conclusions. However, there has been various studies evaluating competency of this important skill among nurses and does not add to the literature. Previous studies including systematic review by Chen et al have demonstrated similar results (PMID: 34989423).
One way to improve the manuscript and present new data would be to analyze different solutions to the current known problem of limited competency in electrocardiogram interpretation. A prospective study of different methods of education would add more to literature.
Further minor points:
1. The introduction is too long and needs to be more concise. Focusing on what is currently known, what is not and how this study addresses the topic.
2. Results . Line 202 - "Some 81.2%" is incorrect usage. "Most (30.25). Would report the numbers and avoid adjectives/descriptors prior in results.
3. in discussion, Line 315 "Non-Saudi nurses scored 315 slightly higher than Saudi nurses. Although this finding is difficult to explain, it may be 316 attributed to the fact there were 163 non-Saudi participants in this study and 92 Saudi 317 participants. " - this sentence does not make sense. How does the country of the participant affect ECG interpretation
Reviewer 3 Report
Congratulations for this reserch
I have a few suggestions:
The title: "Competency in ECG Interpretation and Arrhythmias...."
This mentions the interpretation of the ECG by nurses...
But the development of the research is about RECOGNITION of arrhythmias by ICU and UCC nurses
I suggest modifying the title: Eliminate the word "Interpretation" and replace it with " recognition "
Example: ECG arrhythmia recognition competence and management, among intensive care nurses in Saudi Arabia: a cross-sectional study
Because ECG interpretation consists of recognizing and describing cavity growth, repolarization, direction and direction of P and T waves, QRS axis, etc. among other components of the ECG.
In methodology, could you explain if the ECGs were all made with the same technique (tracing speed, ECG voltage, same paper) Since technical variations in the taking influence the recognition of arrhythmias.
Round 2
Reviewer 2 Report
The authors have addressed the comments and is acceptable for publication.